# Endocrine Mucin-Producing Sweat Gland Carcinoma (EMPSGC) in a Dog: Immunohistochemical Characterization

**DOI:** 10.3390/ani14243637

**Published:** 2024-12-17

**Authors:** Warisraporn Tangchang, Gi-young Jung, Jun-yeop Song, Poornima Kumbukgahadeniya, Dae-hyun Kim, Hyo-jung Kwon, Hwa-young Son

**Affiliations:** 1College of Veterinary Medicine, Chungnam National University, Daejeon 34134, Republic of Korea; waris0770@gmail.com (W.T.); lakshini93@gmail.com (P.K.); hyojung@cnu.ac.kr (H.-j.K.); 2JUNG Animal Clinic, Daejeon 35015, Republic of Korea; jgy8866@hanmail.net

**Keywords:** canine, CK19, endocrine mucin-producing sweat gland carcinoma, Sox9, synaptophysin

## Abstract

A seven-year-old dog developed several small, oval-shaped masses on its abdominal skin. These masses were surgically removed, and further examination under a microscope revealed characteristics similar to certain rare types of cancers, including sweat gland and mammary gland cancers. This study aimed to determine whether the tumor was a sweat gland carcinoma (SGC) or a mammary gland carcinoma (MGC) using specific markers. The tumor tested positive for markers typically found in neuroendocrine cells but negative for the estrogen receptor, which helped confirm that it was a rare type of sweat gland cancer called endocrine mucin-producing sweat gland carcinoma (EMPSGC). This is the first detailed report of this rare tumor in dogs, providing valuable information for veterinarians and researchers who may encounter similar cases.

## 1. Introduction

Mammary and sweat glands originate from the ectoderm during embryonic development [1], which explains their similar structure and function [2]. Consequently, distinguishing between tumors arising from these glands can be challenging due to their morphological and clinical similarities [2,3]. While both mammary gland carcinoma (MGC) and sweat gland carcinoma (SGC) share overlapping histological and immunohistochemical features [2,4], their treatment and prognosis differ significantly. Thus, accurate differentiation is essential for proper clinical management.

The World Health Organization’s (WHO) fourth edition of the Classification of Skin Tumors has emphasized the importance of immunohistochemical (IHC) analysis in identifying adnexal tumors, such as sweat gland neoplasms [5]. Various cytokeratin (CK) markers, including CK5 [4], CK8 [6], CK18 [6], CK8/18 [7], and CK19 [6,8], are routinely used to diagnose skin appendage tumors, particularly those of sweat gland origin. Additionally, Sox9, a marker involved in the development of various tissues, including skin adnexa, is instrumental in distinguishing between sweat and mammary gland neoplasms [9,10,11]. The IHC markers CK5, vimentin, and p63 are markers of myoepithelial cells (MECs) in both sweat and mammary glands [7,12], and they have been used to identify tumor characteristics [7]. The expression of E-cadherin has been studied in both breast [13] and sweat gland cancers [7], as it is an epithelial marker that is useful for differentiation and prognosis in cancer [13].

Endocrine mucin-producing sweat gland carcinoma (EMPSGC) is a low-grade adnexal tumor with neuroendocrine differentiation [14,15,16]. This tumor, originally described in human cases, is rare, and information on animal cases is limited. EMPSGC shares histological characteristics with other neuroendocrine tumors, such as solid papillary carcinoma and ductal carcinoma in situ of the mammary gland [15,16]. Histologically, EMPSGC exhibits mucin accumulation, neuroendocrine features [15], and poor nuclear differentiation. From an IHC perspective, EMPSGC shows immunoreactivity for endocrine markers such as synaptophysin and chromogranin [17].

This study reports the first canine case of EMPSGC, focusing on its diagnostic challenges and the utility of immunohistochemical markers in distinguishing it from other adnexal tumors.

## 2. Materials and Methods

### 2.1. Histopathology and Immunohistochemistry

#### 2.1.1. Histological Evaluation

Tissues were fixed in 10% neutral-buffered formalin, embedded in paraffin, and sectioned at 4 µm thickness for histopathological examination. Sections were stained with hematoxylin and eosin (H&E) and periodic acid–Schiff (PAS) stains. The diagnosis and tumor classification were made based on distinguishing the morphological characteristics. For confirm EMPSGC, essential features include a low-grade, non-metastatic adenocarcinoma of sweat gland origin, with solid, papillary, and/or cribriform architecture, and mucin-filled intracellular or extracellular areas as the key features [18]. Digital images were captured using an Olympus BX51 microscope (Olympus Corporation, Tokyo, Japan) and image transfer software (Dixi eXcope version 1.0.0.48).

#### 2.1.2. Immunohistochemistry

Immunohistochemistry was performed using antibodies for CK19, Sox9, CK5, AE1/AE3 + CK8/18, p63, vimentin, E-cadherin, and synaptophysin to confirm and enhance the accuracy of the diagnosis. Information regarding each antibody is summarized in Table 1. Immunohistochemistry was carried out on deparaffinized sections using the Avidin–Biotin Complex (ABC) kit (VECTASTAIN^®^ Elite^®^ ABC Universal Kit Peroxidase, Vector Laboratories^®^, Burlingame, CA, USA). High-temperature antigen retrieval (HTAR) was performed using citrate buffer (pH 6.0) to unmask the antigens. Sections were incubated with 3% hydrogen peroxide in distilled water for 15 min to quench the endogenous peroxidase activity. The sections were then incubated in a humidified chamber at room temperature for 120 min with diluted normal blocking serum. Afterward, they were incubated with primary antibodies, diluted in normal horse serum (for mouse antibodies) or goat serum (for rabbit antibodies), overnight at 4 °C. Following incubation with primary antibodies, slides were washed with phosphate-buffered saline (PBS) and incubated with the appropriate biotinylated secondary antibody at room temperature for 120 min. Sections were then incubated for 30 min with ABC reagent. The slides were developed with a chromogen solution containing 3.3′-diaminobenzidine tetrachloride (DAB Peroxidase Substrate Kit, Vector Laboratories^®^, Burlingame, CA, USA), and the nuclei were counterstained with hematoxylin. Control slides were performed by excluding the primary antibody as a negative control. Normal skin cell components from two cases were used as internal positive controls, except for synaptophysin, which was compared with the expression in the islets of Langerhans. The intensity of the nuclear or cytoplasmic staining was scored semi-quantitatively on a scale of 0 to 3 using the 10× objective, as follows: 0 (absent), 1 (weak), 2 (moderate), and 3 (strong) [19]. Cell membrane reactivity was graded from 0 to 3 based on the extent of membrane staining: 0 (absent), 1 (weak) for incomplete membrane staining visible at 40×, 2 (moderate) for 20×, and 3 (strong) for 10× objective [20]. The extent of immunoreactivity was evaluated according to the percentage of positive tumor cells: 0 (no staining), 1 (<5%), 2 (5–25%), 3 (26–50%), 4 (51–75%), and 5 (76–100%) [19].

### 2.2. Case Presentation

#### 2.2.1. Case 1

A seven-year-old spayed female poodle presented with several small, oval-shaped nodules approximately the size of rice grains on the skin near the left second mammary gland (Figure 1a). Despite the presence of these nodules, the dog’s appetite, water intake, urine output, and overall vitality remained within the normal limits. During physical examination, a slight inspiratory bronchial sound was noted upon pulmonary auscultation, while both cardiac auscultation and heartworm testing were normal. A comprehensive diagnostic evaluation, including blood tests (complete blood count, serum chemistry panels, C-reactive protein, symmetrical dimethylarginine, and thyroxine levels) and thoracic radiographs, revealed no significant abnormalities.

One month after the initial presentation, the skin overlying the abdominal masses became bluish in color, and the nodules coalesced into a larger, irregular mass. The mass was firm and caused a visible protrusion of the skin. Surgical excision was performed at JUNG Animal Clinic, during which it was noted that the mass was firmly adherent to the underlying skin and subcutaneous connective tissue and exhibited moderate bleeding (Figure 1b). The mass was successfully removed, and the postoperative recovery was uneventful. The dog’s prognosis was favorable, with no signs of recurrence observed to date.

#### 2.2.2. Case 2

A sixteen-year-old neutered female Maltese presented with a 7 cm × 6 cm mass on the right 2nd mammary gland, persisting for five years. The mass was excised (Figure 1c) at the Chungnam National University-Veterinary Medicine Teaching Hospital, and no metastatic lesions were detected.

## 3. Results

### 3.1. Case 1

Microscopically, the tumor was organized into lobules divided by thick fibrous connective stroma. The tumor exhibited a heterogeneous architectural pattern, including solid nests, *pseudoglandular*, and cystic formations filled with mucinous material (Figure 2a). The round to oval neoplastic cells had eosinophilic cytoplasm with pleomorphic nuclei, and occasional mitotic figures were observed (Figure 2b). Focal areas of mucin production and deposits were confirmed by PAS staining (Appendix A). There was no evidence of necrosis, although some inflammatory cells were identified in the interstitial areas. The immunohistochemical analysis (Table 2) showed positive staining for CK5, AE1/AE3+CK8/18, and CK19 (Figure 3a–c), p63, vimentin, and Sox9 (Figure 4a–c), E-cadherin, and synaptophysin, while the estrogen receptor staining was negative (Figure 5a–c). The presence of neuroendocrine differentiation, confirmed by positive synaptophysin staining, along with the overall histomorphological features, led to the final diagnosis of EMPSGC.

### 3.2. Case 2

The tumor from the sixteen-year-old Maltese was histologically diagnosed as MGC. Microscopically, the tumor showed poorly defined margins (Figure 2c) with two mitotic counts per 10 high-powered fields, and no lymphovascular invasion was observed (Figure 2d). The immunohistochemistry revealed positive expressions of CK5 (Figure 3d), E-cadherin (Figure 5d), and p63 and vimentin in the myoepithelial cells (Figure 4d,e), while markers such as AE1/AE3+CK8/18 (Figure 3e), CK19 (Figure 3f), and Sox9 (Figure 4f) were negative (Figure 5d). Estrogen and synaptophysin immunoreactivities were positive (Figure 5e,f), supporting the diagnosis of MGC.

## 4. Discussion

The diagnosis of EMPSGC presents a significant challenge in human pathology due to its rarity and histological similarities to other adnexal tumors, particularly MGC [14,15,16,17,21]. Accurate differentiation is critical because the treatment and prognosis of SGC and MGC differ significantly [4]. This study reports the first documented case of EMPSGC in a dog, highlighting the use of IHC markers to distinguish [22] between these tumors and offering insights into the clinical management of similar cases in veterinary practice. For the EMPSGC markers, data from human studies were compared with our canine case. These markers were used to confirm their relevance to the canine case by utilizing antibodies previously validated for immunohistochemistry in this species to minimize cross-reactivity. Additionally, both positive and negative controls were included in this study to detect false-negative and false-positive results, thereby supporting the validation of species-specific assays [23].

The IHC marker Sox9 was pivotal in this case, allowing for a clear differentiation between SGC and MGC, thereby ensuring accurate diagnosis and proper clinical management. For SGC, a combination of CK19, a well-known marker for adnexal tumors [6,7,8], and Sox9 [9,10,11], a transcription factor involved in glandular differentiation, expression played a pivotal role in confirming the sweat gland origin of the tumor in case 1. There was great sensitivity and specificity with SGC [11], and the immunoreactivity pattern was nuclear and mostly located in basaloid cells of the tumor nest [9]; this is similar to our sweat gland case, where palisading cells were readily positive for SOX9, while the mammary gland showed a negative result. This finding is essential given the overlapping histological features of SGC and MGC, particularly in tumors located near mammary tissue.

Previous studies have shown that the combination of CK5 and p63 is a reliable indicator for confirming SGC, distinguishing it from MGC, and is frequently observed in SGC [2,4]. p63 is typically nuclear-positive in less malignant and primary tumors, and its expression is primarily detected in the benign forms of both SGC and MGC [24,25]. However, p63 expression is reduced or absent in more invasive tumors due to the transformation and loss of myoepithelial cell (MEC) characteristics [2,26]. Immunohistochemically, the myoepithelial markers commonly used in sweat gland tumors include CK5, vimentin, and p63. The immunohistochemical staining patterns reflecting the muscular–epithelial characteristics of MECs hold significant diagnostic value in determining the origin of tumors [7].

Although previous studies have reported negative results for E-cadherin in sweat gland carcinoma [13], the presence of both vimentin and E-cadherin in EMPSGC suggests a complex biological state, likely indicating that the tumor cells are in a hybrid or partial epithelial–mesenchymal transition (EMT) phase. This co-expression reflects tumor cells transitioning between epithelial and mesenchymal phenotypes, conferring greater plasticity and potentially enhancing their ability to invade locally or progress [27]. In our case, the observed immunoreactivity appears to correlate with subtypes that are associated with a good prognosis [13]. Similarly, the breast neoplastic cells in our study are undergoing a transition that weakens their epithelial adhesion, but they have not yet fully adopted mesenchymal traits characteristic of an invasive phenotype. This incomplete EMT may explain why these cells have not yet reached full invasiveness [27,28]. Additionally, previous studies have employed E-cadherin as a diagnostic marker to distinguish between ductal and lobular mammary carcinomas [20].

In addition to CK19 and Sox9, synaptophysin staining revealed neuroendocrine differentiation [14,15,25], a hallmark of EMPSGC, further supporting the diagnosis. Neuroendocrine differentiation is unusual in typical sweat gland tumors, making this case unique. Previous studies have reported synaptophysin as an inconsistent neuroendocrine marker due to its variable expression intensity [17,29,30]. Despite this variability, the observed synaptophysin expression highlights its potential utility as a supportive marker for this tumor. The absence of estrogen receptor (ER) expression in case 1, which is often positive in mammary gland tumors, further confirmed the sweat gland origin of the tumor. These findings are consistent with human cases of EMPSGC, although species-specific differences may exist, such as the variation in hormonal receptor expression.

This case underscores the importance of employing a comprehensive IHC panel to distinguish between SGC and MGC. Misdiagnosis can lead to inappropriate treatment [4] and negatively impact patient outcomes, as these tumors have different therapeutic protocols [12]. The IHC panel used in this study, including markers such as CK19, Sox9, p63, and synaptophysin, provided a reliable method for differentiating between these neoplasms and offered crucial diagnostic clarity.

From a clinical perspective, the surgical excision of the tumor in case 1 resulted in a favorable outcome, with no signs of recurrence during follow-up. This suggests that complete surgical removal may be an effective treatment [22,31] for EMPSGC in dogs. While recurrence has been reported in human cases of EMPSGC [31], the absence of recurrence in this canine case is encouraging, though long-term follow-up remains essential to fully understand the tumor’s behavior in animals. Given the limited data on EMPSGC in veterinary medicine, future studies are necessary to investigate the long-term prognosis and potential for metastasis or recurrence in similar cases.

This case also contributes to the broader understanding of adnexal tumors in veterinary oncology. As the World Health Organization (WHO) continues to update the Classification of Skin Tumors, particularly in the fourth edition, this report supports the need for detailed histopathological and immunohistochemical evaluations of skin tumors in animals [5]. The identification of reliable IHC markers such as CK19 and Sox9 adds to the diagnostic toolkit for veterinarians, allowing for a more accurate differentiation of skin adnexal tumors, including rare neoplasms like EMPSGC.

In conclusion, this report provides valuable insights into the diagnosis and clinical management of EMPSGC in dogs, representing the first documented case in the veterinary literature. By highlighting the diagnostic utility of IHC markers such as CK19, Sox9, and synaptophysin, this study offers a framework for the accurate diagnosis of rare adnexal tumors. The findings underscore the importance of a multidisciplinary diagnostic approach and contribute to the growing body of knowledge regarding the behavior and management of EMPSGC in animals. Future research should focus on the biological behavior and optimal treatment strategies for EMPSGC, as well as a further exploration of its occurrence in other species.

## 5. Conclusions

This study documents the first case of endocrine mucin-producing sweat gland carcinoma (EMPSGC) in a dog, successfully differentiating it from other apocrine gland tumors and MGC using immunohistochemical markers such as CK19, Sox9, CK5, p63, and vimentin. Accurate diagnosis is crucial due to the differing treatment strategies used for these tumors. Surgical excision led to a favorable outcome with no recurrence observed. While rare, EMPSGC requires further study to understand its long-term behavior and optimal treatment in veterinary medicine.

## Figures and Tables

**Figure 1 animals-14-03637-f001:**
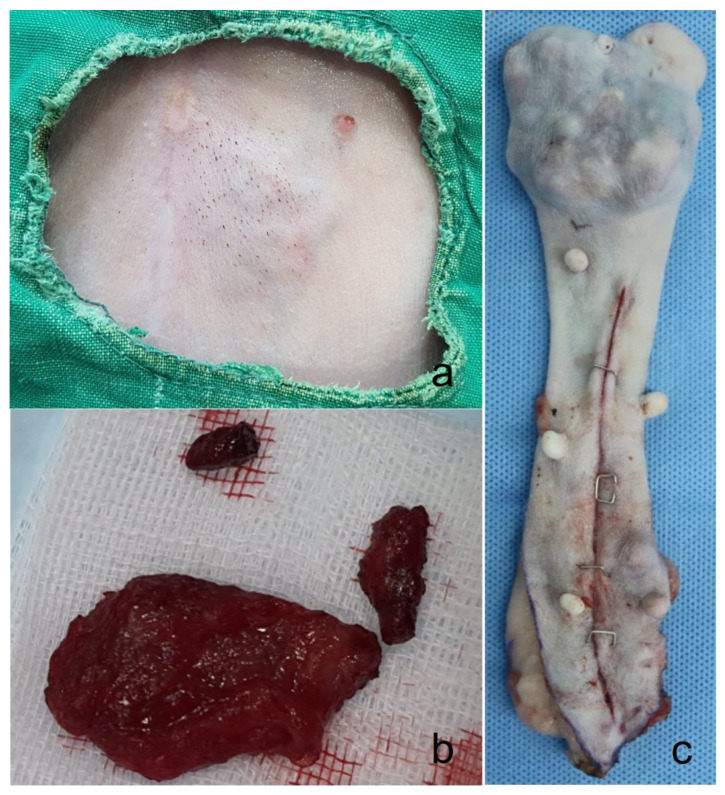
Gross finding figure. (**a**) The skin shows bluish discoloration, and numerous nodules form irregular mass that protrude into the skin of case 1. (**b**) Removed tumor mass from case 1; the mass is firmly adhered to the skin and subcutaneous connective tissue. (**c**) A removed mass (7 cm × 6 cm) in the right 2nd mammary gland from case 2.

**Figure 2 animals-14-03637-f002:**
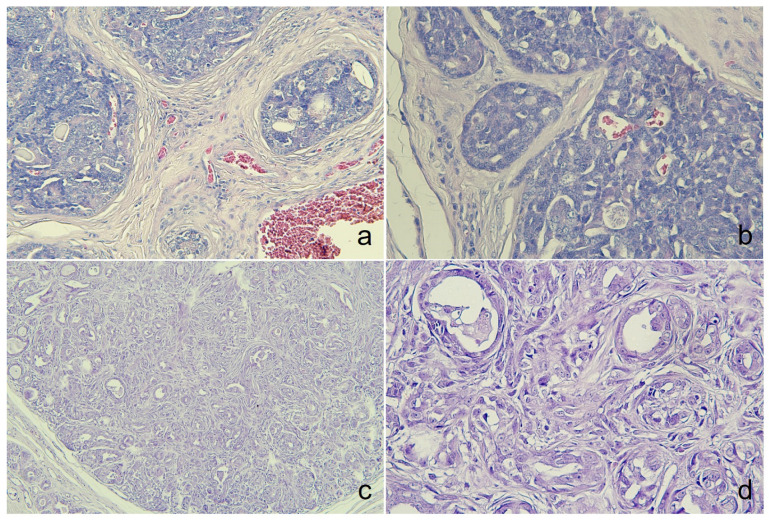
Histological examination of both cases. (**a**) The tumor displayed a thickened stroma of lobular. The multilobular tumor exhibits mucin secretion in the cystic and solid areas, as observed in case 1, Mag. = ×200. (**b**) Within the lobules, peripheral palisading was identified in some areas. Nuclei were bland with moderate pleomorphism and diffusely stippled chromatin from case 1, Mag. = ×400. (**c**) The irregular proliferation of small glands of case 2, Mag. = ×100. (**d**) The tubules were lined by a single layer of cuboidal or columnar cell hyperplasia with atypia of case 2, Mag. = ×400, hematoxylin and eosin (HE).

**Figure 3 animals-14-03637-f003:**
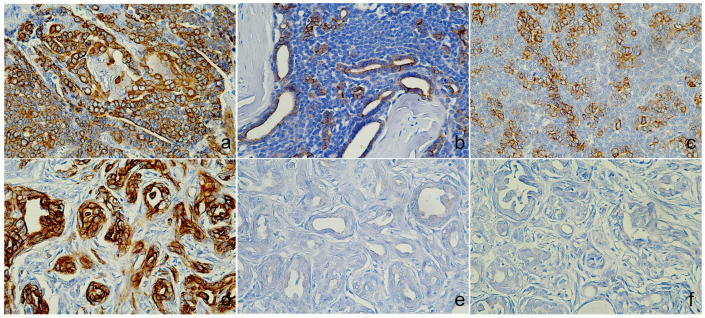
Immunoreactivity of cytokeratin (CK). Case 1 (sweat gland carcinoma), (**a**) luminobasal tumor cells were positive for CK5, (**b**) luminal tumor cells were positive for AE1/AE3+CK8/18, (**c**) CK19 was cytoplasmic, and cell membrane expression. Case 2 (mammary gland carcinoma), (**d**) CK5 was positive, (**e**) AE1/AE3+CK8/18 and (**f**) CK19 showed no immunoreactivity, immunohistochemistry (IHC).

**Figure 4 animals-14-03637-f004:**
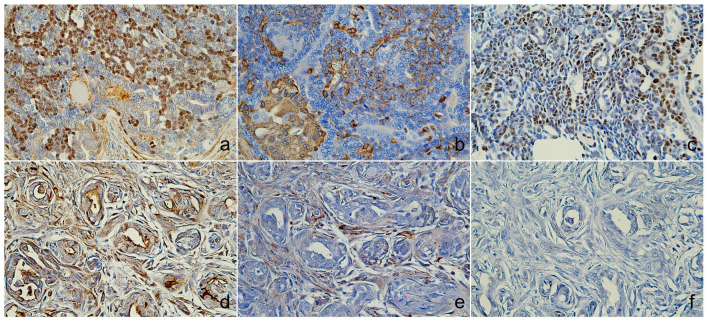
Immunoreactivity of myoepithelial and other markers. Case 1 (sweat gland carcinoma), (**a**) nuclear was positive for p63, (**b**) cytoplasmic areas were positive for vimentin, (**c**) Sox9 found nuclear expression. Case 2 (mammary gland carcinoma), (**d**) p63 and (**e**) vimentin were positive in myoepithelial cells, and (**f**) Sox9 was negative expression, immunohistochemistry (IHC).

**Figure 5 animals-14-03637-f005:**
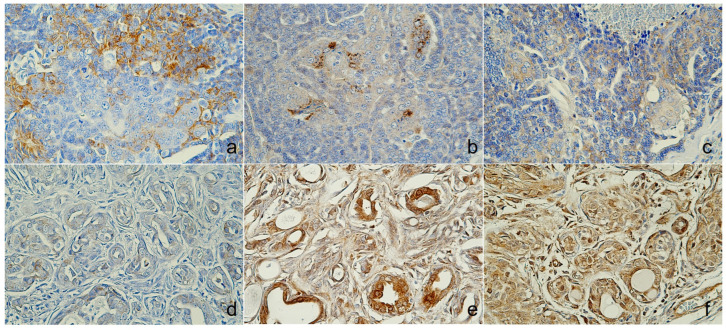
Immunoreactivity of endocrine and other markers. Case 1 (sweat gland carcinoma), (**a**) membranous and cytoplasmic areas were positive for E-cadherins, (**b**) cytoplasmic expression of synaptophysin, (**c**) nuclear was negative for estrogen. Case 2 (mammary gland carcinoma), (**d**) E-cadherins showed weak expression, (**e**) synaptophysin was cytoplasmic expression, and (**f**) nuclear cell showed positive for estrogen, immunohistochemistry (IHC).

**Table 1 animals-14-03637-t001:** Immunohistochemistry antibody information.

Antibody	Clonality ^1^	Clone	Species	Source of Antibody	Dilution
CK5	mAb	ab52635	rabbit	Abcam (Cambridge, UK)	1:200
AE1/3+CK8/18 ^2^	pAb	ab86734	mouse		1:250
p63	mAb	ab124762	rabbit		1:1000
Vimentin ^2^	mAb	MAB3400	mouse	Sigma-Aldrich (St. Louis, MO, USA)	1:200
Sox9 ^3^	mAb	ab185966	rabbit	Abcam (Cambridge, UK)	1:1000
Synaptophysin	mAb	ab32127	rabbit		1:100
Estrogen	mAb	ab32063	rabbit		1:250
E-cadherin	mAb	4A2	mouse	Cell Signaling Technology (Danvers, MA, USA)	1:200
CK19	mAb	D7F7W	mouse		1:100

^1^ Monoclonal antibody = mAb, polyclonal antibody = pAb, ^2^ reactivity in canine, ^3^ predicted reactivity in canine.

**Table 2 animals-14-03637-t002:** Comparison of immunohistochemical (IHC) results between sweat gland carcinoma (SGC; case 1) and mammary gland carcinoma (MGC; case 2).

IHC Marker		SGC								MGC							
		Intensity				Extent				Intensity				Extent			
Name	Area	B	L	M	S	B	L	M	S	B	L	M	S	B	L	M	S
CK5	Cy	2	3	0	0	2	2	0	0	3	4	0	0	5	3	0	0
p63	N	2	0	3	0	3	0	4	0	0	0	1	0	0	0	1	0
Vimentin	Cy	2	0	3	0	3	0	4	0	0	0	1	0	0	0	1	0
AE1/3+CK8/18	C, Cy	0	2	0	0	0	3	0	0	0	0	0	0	0	0	0	0
E-cadherin	C	3	2	0	0	3	2	0	0	1	1	0	0	3	3	0	0
Synaptophysin	Cy	0	0	0	3	0	0	0	3	3	3	0	0	1	1	0	0
Estrogen	N	0	0	0	0	0	0	0	0	1	0	0	0	3	0	0	0
CK19	C, Cy	2	2	0	0	3	4	0	0	0	0	0	0	0	0	0	0
Sox9	N	2	0	3	0	4	0	5	0	0	0	0	0	0	0	0	0

Luminal = L, basal = B, sebaceous-like = S, myoepithelial = M, cell membrane = C, cytoplasm = Cy, nucleus = N, intensity; 0 = absent, 1 = weak, 2 = moderate, and 3 = strong extent; 0 = no staining; 1 = <5%; 2 = 5–25%; 3 = 26–50%; 4 = 51–75%; and 5 = 76–100% of positive tumor cells.

## Data Availability

Data are contained within the article.

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
