# Peer review of "Endocrine Mucin-Producing Sweat Gland Carcinoma (EMPSGC) in a Dog: Immunohistochemical Characterization"

_animals, 2024, doi:10.3390/ani14243637_

Round 1
Reviewer 1 Report
Comments and Suggestions for Authors
Dear authors, thank you for submit this manuscript
Some comments and suggestions:
Line 10: “spindle shaped masses”. Please define what this means macroscopically. As a rule, the definition spindle shaped is attributed to cells during histological evaluation and not during macroscopic evaluation.
Introduction:
The authors' introduction is too brief and some things are left unexplained.
For example, the authors only mention the markers SOX and CK. There is nothing to explain the relevance of vimentin and E-cadherin in these tumors or their relevance to this case report.
Furthermore, it is not contradictory that some tumors are positive for vimentin and also for e-cadherin, since they usually have an inverse relationship. Vimentin is a mesenchymal marker and e-cadherin epithelial. In a situation of epithelial-mesenchymal transition, they even have an inverse relationship. Please explain this.
Materials and Methods:
There are serious flaws here.
The authors should separate the histological evaluation from the immunohistochemistry. The authors do not mention any histological features that were evaluated in the materials and methods. There should be a detailed explanation of everything that was evaluated. What points to malignancy, what helps in the diagnosis, etc.
The same goes for immunohistochemistry. The authors only make a brief reference to the protocol which is, in fact, insufficient. The whole protocol should be explained in detail. Furthermore, what are the criteria for evaluating the immunostain? What is a positive immunostain for the authors? What is the extent of labeling (% of labeled cells)? What is the intensity of labeling? What are the positive and negative immune controls?
Furthermore, are the biomarkers chosen already validated for the dog? If not, how have the specificity and cross-reactivity of the markers been assessed?
Table 1: change the word multiclonal to polyclonal
Discussion:
Same problem. The authors only discuss SOX and CK and make no reference to the other markers
Although we know that this disease is rare, the number of references cited seems low.
Author Response
Reviewer 1
Comments 1: Line 10: “spindle shaped masses”. Please define what this means macroscopically. As a rule, the definition spindle shaped is attributed to cells during histological evaluation and not during macroscopic evaluation.
Response 1: Thank you for pointing this out. I agree with this comment. Therefore, I have defined “oval shaped masses” instead of “spindle shaped masses” in page number 1 to 2 of 9, and line 10, 19, 102
Comments 2: Introduction:
The authors' introduction is too brief and some things are left unexplained. For example, the authors only mention the markers SOX and CK. There is nothing to explain the relevance of vimentin and E-cadherin in these tumors or their relevance to this case report. Furthermore, it is not contradictory that some tumors are positive for vimentin and also for e-cadherin, since they usually have an inverse relationship. Vimentin is a mesenchymal marker and e-cadherin epithelial. In a situation of epithelial-mesenchymal transition, they even have an inverse relationship. Please explain this.
Response 2: Agree. I have, accordingly, revised vimentin, E-cadherin, and p63 and synaptophysin to emphasize this point. This change can be found – page number 2 of 9, and line 49-53 and 60-61]
Comments 3: Materials and Methods:
There are serious flaws here.
The authors should separate the histological evaluation from the immunohistochemistry. The authors do not mention any histological features that were evaluated in the materials and methods. There should be a detailed explanation of everything that was evaluated. What points to malignancy, what helps in the diagnosis, etc.
The same goes for immunohistochemistry. The authors only make a brief reference to the protocol which is, in fact, insufficient. The whole protocol should be explained in detail. Furthermore, what are the criteria for evaluating the immunostain? What is a positive immunostain for the authors? What is the extent of labeling (% of labeled cells)? What is the intensity of labeling? What are the positive and negative immune controls?
Furthermore, are the biomarkers chosen already validated for the dog? If not, how have the specificity and cross-reactivity of the markers been assessed?
Response 3: Thank you for pointing this out. I agree with this comment. Therefore, I have separated the histological evaluation from the immunohistochemistry and detailed explanation about points of malignancy and diagnosis. For immunohistochemistry, I explained in detail of protocol and added information about positive and negative control in materials and methods but unfortunately, we didn’t evaluate an intensity and frequency (%) of expression. This change can be found – page number 2 and 3 of 9, and line 66-99
Comments 4: Table 1: change the word multiclonal to polyclonal
Response 4: Agree. I have, accordingly, changed multiclonal to polyclonal. this change can be found on table 1– page number 3 of 9
Comments 5: Discussion:
Same problem. The authors only discuss SOX and CK and make no reference to the other markers
Response 5: Agree. I have, accordingly, added more references to the other markers to emphasize this point. This change can be found – page number 7 of 9, and line 202-209
Comments 6: Although we know that this disease is rare, the number of references cited seems low.
Response 6: Agree. I have, accordingly, added more references to the other markers to emphasize this point as follows;
5. Elder, D.; Massi, D.; Scolyer, A.; Willemze, R. WHO Classification of Skin Tumours; IARC: Lyon, France, 2018; pp. 168–169.
12. Gusterson BA, Stein T. Human breast development. Semin Cell Dev Biol. 2012, 23(5):567-573.
13. Chintamani, Rekhi B, Bansal A, Bhatnagar D, Saxena S. Expression of E-Cadherin in breast carcinomas and its association with other biological markers - a prospective study. Indian J Surg Oncol. 2010, 1(1):40-46.
17. Cazzato G, Bellitti E, Trilli I, et al. Endocrine Mucin-Producing Sweat Gland Carcinoma: Case Presentation with a Comprehensive Review of the Literature. Dermatopathology (Basel). 2023, 10(3):266-280.
20. Ivan D, Hafeez Diwan A, Prieto VG. Expression of p63 in primary cutaneous adnexal neoplasms and adenocarcinoma metastatic to the skin. Mod Pathol. 2005,18(1):137-142.
22. Gama A, Alves A, Gartner F, Schmitt F. p63: a novel myoepithelial cell marker in canine mammary tissues. Vet Pathol. 2003, 40(4):412-420.
23. Acs G, Lawton TJ, Rebbeck TR, LiVolsi VA, Zhang PJ. Differential expression of E-cadherin in lobular and ductal neoplasms of the breast and its biologic and diagnostic implications. Am J Clin Pathol. 2001, 115(1):85-98.

Reviewer 2 Report
Comments and Suggestions for Authors
The authors present what they claim to be the first case of an endocrine mucin-producing sweat gland carcinoma (EMPSGC) in a dog. While this topic is novel and potentially valuable to the field of veterinary pathology, such a groundbreaking finding requires more thorough and convincing evidence to support the diagnosis. Below, I outline several major and minor concerns that, in my view, need to be addressed to strengthen the validity and impact of the study.
Major Concerns:
1. Insufficient Evidence for EMPSGC Diagnosis
To differentiate EMPSGC from general apocrine gland tumors, three hallmark features must be demonstrated:
- Neuroendocrine differentiation: This requires convincing immunohistochemical evidence.
- Solid, papillary, and/or cribriform architecture: These structures must be clearly identifiable histologically.
- Intracellular or extracellular mucin: This should be evident through routine or special staining.
The histological evidence provided in Figure 2A and B is unconvincing:
The H&E images lack proper staining quality, as eosin staining is minimal, appearing as if only hematoxylin counterstaining was performed. The authors describe pseudorosette, cribriform architecture, and cystic formation with mucinous material in Figure 2A. However, the low magnification makes it difficult to evaluate these features, and based on the provided images, the lesion appears more consistent with a typical apocrine ductal carcinoma. To strengthen their claim, the authors should provide properly stained H&E images at an appropriate magnification. Special stains such as PAS or Alcian blue should be applied to confirm the presence of mucin. Additionally, they should refer to similar figures in cited references (e.g., references 12 and 16) to ensure their findings are comparable.
2. Immunohistochemical Characterization
Immunohistochemical (IHC) findings raise significant questions regarding both the specificity of the antibodies and the sufficiency of the evidence:
- Neuroendocrine markers: Only synaptophysin is tested as a neuroendocrine marker. For a first report of EMPSGC, this is insufficient. Additional markers such as chromogranin or neuron-specific enolase (NSE) are recommended to confirm neuroendocrine differentiation. There is a concerning observation in Figure 5e: the neuroendocrine marker synaptophysin is also positive in basal and luminal cells of mammary gland carcinoma. This raises questions about the antibody’s specificity, as such findings are unusual for mammary epithelial cells. Negative and positive controls should be included to validate antibody specificity. Furthermore, Figure 5b shows weak synaptophysin expression, which further weakens the case for EMPSGC.
- Other markers: E-cadherin is discussed but raises concerns, as it is unexpectedly negative in Figure 5d. This result is unusual, particularly in a mammary tumor that does not appear highly malignant based on its morphology. E-cadherin is typically expressed in luminal and basal epithelial cells of mammary gland tumors, and its absence would be more consistent with a highly invasive phenotype, which is not evident in this case
3. CK19 as a Key Marker for SGC vs. MGC
The authors claim CK19 is pivotal for distinguishing SGC from MGC. However, the reported absence of CK19 in MGC is problematic: CK19 is a marker typically expressed in luminal epithelial cells, including glandular epithelium in mammary ducts. Its absence is unusual, especially in low-grade mammary carcinomas as implied by Figure 2. Relevant literature (e.g., Noguchi et al., 2019; Gama et al., 2010) indicates CK19 expression is common in mammary gland tumors. The authors should discuss this inconsistency and provide a more thorough comparison.
Minor Concerns:
- Inconsistent Citations:
Line 42: If referencing the WHO 4th edition, the authors should cite the original book rather than review articles. Line 155: The claim that EMPSGC poses a diagnostic challenge in veterinary pathology is unsupported by veterinary literature and relies solely on human medical references.
Conclusion:
The manuscript raises an interesting hypothesis by presenting what could be the first case of EMPSGC in a dog. However, the provided evidence, particularly the histological and immunohistochemical characterization, is insufficient to substantiate this claim. The study requires significant additional work.
Author Response
Reviewer 2
Comments 1: 1. Insufficient Evidence for EMPSGC Diagnosis
To differentiate EMPSGC from general apocrine gland tumors, three hallmark features must be demonstrated:
- Neuroendocrine differentiation: This requires convincing immunohistochemical evidence.
- Solid, papillary, and/or cribriform architecture: These structures must be clearly identifiable histologically.
- Intracellular or extracellular mucin: This should be evident through routine or special staining.
The histological evidence provided in Figure 2A and B is unconvincing:
The H&E images lack proper staining quality, as eosin staining is minimal, appearing as if only hematoxylin counterstaining was performed. The authors describe pseudorosette, cribriform architecture, and cystic formation with mucinous material in Figure 2A. However, the low magnification makes it difficult to evaluate these features, and based on the provided images, the lesion appears more consistent with a typical apocrine ductal carcinoma. To strengthen their claim, the authors should provide properly stained H&E images at an appropriate magnification. Special stains such as PAS or Alcian blue should be applied to confirm the presence of mucin. Additionally, they should refer to similar figures in cited references (e.g., references 12 and 16) to ensure their findings are comparable.
Response 1: Thank you for pointing this out. We agree with this comment. Therefore, I have revised Insufficient Evidence for EMPSGC Diagnosis and reviewer mentioned how to differentiate EMPSGC from general apocrine gland tumors, three hallmark features must be demonstrated:
- Neuroendocrine differentiation: Even we performed one immunohistochemical marker. Some areas from routine histological finding could support this type of differentiation.
- Solid, papillary, and/or cribriform architecture: we performed be clearly identifiable histologically. We provided H&E images with proper staining quality and changed appropriate image which consist of pseudorosette, cribriform architecture, and cystic formation with mucinous material in Figure 2A and B.
- Intracellular or extracellular mucin: We provided PAS images confirm the presence of mucin in Supplement Figure 1.
Comments 2: 2. Immunohistochemical Characterization
Immunohistochemical (IHC) findings raise significant questions regarding both the specificity of the antibodies and the sufficiency of the evidence:
Neuroendocrine markers: Only synaptophysin is tested as a neuroendocrine marker. For the first report of EMPSGC, this is insufficient. Additional markers such as chromogranin or neuron-specific enolase (NSE) are recommended to confirm neuroendocrine differentiation.
Response: Thank you for pointing this out. I agree with this comment. Unfortunately, there are no neuroendocrine markers in our lab, and we have already asked other labs in our school. Moreover, if we order, the delivery period takes more than 3 weeks, so we might not get their expression results within 1 month. Do you have other suggestions for this issue?
There is a concerning observation in Figure 5e: the neuroendocrine marker synaptophysin is also positive in basal and luminal cells of mammary gland carcinoma. This raises questions about the antibody’s specificity, as such findings are unusual for mammary epithelial cells.
Response: Thank you for pointing this out. However, many studies have reported that breast cancer lesions are positive for neuroendocrine (NE) markers, whereas only a small number of studies have reported immunopositivity for NE markers in normal mammary tissues or benign lesions and one paper supported synaptophysin may provide a marker of breast cancer diagnosed by core needle biopsy (Maeda I, et al., 2016) and In canine cases, two of ten cases of solid mammary carcinoma positive for chromogramin A in immunohistochemistry were positive for synaptophysin (Nakagaki KYR, et al., 2021). However, synaptophysin supported NE expression in our SGC for EMPSGC diagnosis.
Negative and positive controls should be included to validate antibody specificity.
Response: Agree. I have, accordingly, added controls information to emphasize this point. Controls were performed that exempt the primary antibody as negative control and normal cell components in the skin of 2 cases themselves were used as internal positive control except synaptophysin which was compared with the expression of the islet of Langerhans. Evaluate the expression status as + express, - non-express and detail the location of the expression.
Other markers: E-cadherin is discussed but raises concerns, as it is unexpectedly negative in Figure 5d. This result is unusual, particularly in a mammary tumor that does not appear highly malignant based on its morphology. E-cadherin is typically expressed in luminal and basal epithelial cells of mammary gland tumors, and its absence would be more consistent with a highly invasive phenotype, which is not evident in this case.
Response 2: Thank you for pointing this out. We agree with this comment. Therefore, we have modified the E-cadherin expression expressed in luminal and basal epithelial cells of mammary gland tumors in table 2
Comments 3:
The authors claim CK19 is pivotal for distinguishing SGC from MGC. However, the reported absence of CK19 in MGC is problematic: CK19 is a marker typically expressed in luminal epithelial cells, including glandular epithelium in mammary ducts. Its absence is unusual, especially in low-grade mammary carcinomas as implied by Figure 2. Relevant literature (e.g., Noguchi et al., 2019; Gama et al., 2010) indicates CK19 expression is common in mammary gland tumors. The authors should discuss this inconsistency and provide a more thorough comparison.
Response 3: Thank you for pointing this out. We agree with this comment. Although CK19 is a marker typically expressed in luminal epithelial cells, there were no expression in both mammary ducts and our low-grade mammary carcinomas. Based on relevant literature and our expression results. We modified the discussion part and claimed only Sox9 as distinguishing marker between MGC and SGC in this study. However, a combination of CK19, Sox9 and other markers expression could be supported SGC diagnosis.
Comments 4:
1.Inconsistent Citations:
Line 42: If referencing the WHO 4th edition, the authors should cite the original book rather than review articles. Line 155: The claim that EMPSGC poses a diagnostic challenge in veterinary pathology is unsupported by veterinary literature and relies solely on human medical references.
Response 4: Thank you for pointing this out. We agree with this comment. Therefore, we have revised and cited a reference as the original book.
Elder, D.; Massi, D.; Scolyer, A.; Willemze, R. WHO Classification of Skin Tumours; IARC: Lyon, France, 2018; pp. 168–169.
Line 155: We adjusted as the diagnosis of EMPSGC presents a significant challenge in human pathology currently on pages number 7 of 9 on lines number 183.

Reviewer 3 Report
Comments and Suggestions for Authors
Dear Authors, the case report is interesting and original, but needs minor revision.
Table n. 1 There are two misaligned overlapping sentences. Please, arrange the table n. 1
2. Materials and methods - Chapter 2.1 Histopathology. In this chapter is described method for histopathology and for immunohistochemistry. Please rename the Chapter 2.1 as follow: Histopathology and Immunoihistochemistry.
Chapter 2.1 Histopathology and Immunohistochemistry
Please, indicate the the temperature of immunoreaction: room temperature or overnight at 4°C?
Please, indicate the preparation of negative immunohistochemistry controls: a) Endogenous tissue background control; b) No primary antibody control.
Please, indicate the nuclei counterstain used in immunohistochemistry
Please, indicate the optical microscope and the imaging capture system used.
Table n. 2 - Please, insert the symbols + and - in captions indicating the meaning
Author Response
Reviewer 3
Comments 1: Table n. 1 There are two misaligned overlapping sentences. Please, arrange the table n. 1
Response 1: Thank you for pointing this out. Unfortunately, I could not find two misaligned overlapping sentences that reviewer mentioned. Could the reviewer guide which line or the sentences?
Comments 2: 2. Materials and methods - Chapter 2.1 Histopathology. In this chapter is described method for histopathology and for immunohistochemistry. Please rename the Chapter 2.1 as follow: Histopathology and Immunoihistochemistry. Chapter 2.1 Histopathology and Immunohistochemistry
Response 2: Agree. I have, accordingly, changed Chapter 2.1 to emphasize this point. This change can be found – page number 2 of 9, and line 66
Comments 3: Please, indicate the temperature of immunoreaction: room temperature or overnight at 4°C?.
Response 3: Thank you for pointing this out. I agree with this comment. Therefore, I have indicated room temperature as the temperature of immunoreaction. This change can be found – page number 2 of 9, and line 83 and 87
Comments 4: Please, indicate the preparation of negative immunohistochemistry controls: a) Endogenous tissue background control; b) No primary antibody control. Please, indicate the nuclei counterstain used in immunohistochemistry
Response 4: Thank you for pointing this out. I agree with this comment. Therefore, I have indicated the preparation of negative immunohistochemistry controls in materials and methods and added control detail in materials and methods This change can be found – page number 2 of 9, and line 83 and 87
Comments 5: Please, indicate the nuclei counterstain used in immunohistochemistry
Response 5: Thank you for pointing this out. I agree with this comment. Therefore, I have indicated the Hematoxylin solution was used for nuclei counterstain in materials and methods. This change can be found – page number 2 and 3 of 9, and line 91-95.
Comments 6: Please, indicate the optical microscope and the imaging capture system used.
Response 6: Agree. I have, accordingly, added optical Olympus BX51 Microscope and Dixi eXcope software to emphasize this point. This change can be found – page number 2 of 9, and line 72-73.
Comments 7: Table n. 2 - Please, insert the symbols + and - in captions indicating the meaning
Response 7: Agree. I have, accordingly, added the meaning of these symbols. This change can be found – page number 5 of 9, and line 166.

Reviewer 4 Report
Comments and Suggestions for Authors
The authors present a case report of an EMPSGC in a dog, and provide IHC characterisation as part of the diagnostic work up. This is a very interesting report and is definitely of interest, as it’s the first detailed description of this rare canine tumor and it also that it provides a useful comparison to mammary gland carcinoma. The report is well-written and clear, and the discussion section is very informative. The figures and tables are excellent. The authors are to be congratulated on their efforts.
My only suggestion would be that it would be that helpful at the Introduction section to let the reader know that even though the title is “Endocrine Mucin-Producing Sweat Gland Carcinoma 2 (EMPSGC) in a Dog”, two tumors will be presented in the Results section. Perhaps also for Table 2, the legend could read “Comparative of Immunohistochemical (IHC) results between sweat gland carcinoma (SGC; Case 1) and mammary gland carcinoma (MGC; Case 2)”, just to help the reader.
Author Response
Comments 1: My only suggestion would be that it would be that helpful at the Introduction section to let the reader know that even though the title is “Endocrine Mucin-Producing Sweat Gland Carcinoma 2 (EMPSGC) in a Dog”, two tumors will be presented in the Results section. Perhaps also for Table 2, the legend could read “Comparative of Immunohistochemical (IHC) results between sweat gland carcinoma (SGC; Case 1) and mammary gland carcinoma (MGC; Case 2)”, just to help the reader.
Response 1: Thank you for pointing this out. We agree with this comment. Therefore, I have added case 1 and 2 on pages number 4 of 8 on lines number 133-134.
Round 2
Reviewer 1 Report
Comments and Suggestions for Authors
Most of the major flaws pointed out in the first review remain. In addition, the text that has been added contains incorrect English that is difficult to understand
Note that in the “histological evaluation” section, the authors have not added the criteria evaluated. They only added the following sentence “The diagnosis 70 and tumor classification based on distinguished morphological characteristics.”
In the immunohistochemistry section, the procedure was well explained, but there is still a serious flaw: how did the authors know that these antibodies were cross-reactive in the canine species? How did they do the validation/testing before the IHC? This needs to be explained
If the authors didn't evaluate the extent of labeling or the intensity of labeling, then what is considered positive labeling? Was there a differentiation between high and low labeling?
This could resolve the question of Vimentin and E-Cadherin which was not answered by the authors. In this manuscript they mention that the same tumor was positive for vimentin and cadherin, but they don't discuss whether, for example, the expression of vimentin was higher and that of cadherin lower (negative correlation) or whether the positivity of each of the biomarkers occurs in different places in the tumor.
Comments on the Quality of English Language
The text added after the first review is difficult to understand.
Author Response
1st Reviewer:
Comments and Suggestions for Authors
- Comment: Most of the major flaws pointed out in the first review remain. In addition, the text that has been added contains incorrect English that is difficult to understand. Note that in the “histological evaluation” section, the authors have not added the criteria evaluated. They only added the following sentence “The diagnosis 70 and tumor classification based on distinguished morphological characteristics.”
Response: Thank you for pointing this out. The diagnosis and tumor classification were made based on distinguishing morphological characteristics. For confirm EMPSGC, essential features include low-grade, non-metastatic adenocarcinoma of sweat gland origin, with solid, papillary, and/or cribriform architecture, and mucin-filled intracellular or extracellular areas as the key features on page 2, lines 70-74.
- Comment: In the immunohistochemistry section, the procedure was well explained, but there is still a serious flaw: how did the authors know that these antibodies were cross-reactive in the canine species? How did they do the validation/testing before the IHC? This needs to be explained
Response: Thank you for pointing this out. I agree with this comment. For the EMPSGC markers, data from human studies were compared with our canine case. These markers were used to confirm their relevance to the canine case by utilizing anti-bodies previously validated for immunohistochemistry in this species to minimize cross-reactivity. Additionally, both positive and negative controls were included in the study to detect false-negative and false-positive results, thereby supporting the validation of species-specific assays on page 7, lines 199-204.
- Comment: If the authors didn't evaluate the extent of labeling or the intensity of labeling, then what is considered positive labeling? Was there a differentiation between high and low labeling?
Response: Agree. Accordingly, I have modified and evaluated the extent and intensity of labeling in Table 2 on page 5. Additionally, the criteria have been explained and references provided on page 3, lines 98-105.
- Comment: This could resolve the question of Vimentin and E-Cadherin which was not answered by the authors. In this manuscript they mention that the same tumor was positive for vimentin and cadherin, but they don't discuss whether, for example, the expression of vimentin was higher and that of cadherin lower (negative correlation) or whether the positivity of each of the biomarkers occurs in different places in the tumor.
Response: Thank you for pointing this out. I agree with this comment. The presence of both vimentin and E-cadherin in EMPSGC suggests a complex biological state, likely indicating that the tumor cells are in a hybrid or partial epithelial-mesenchymal transition (EMT) phase. This co-expression reflects tumor cells transitioning between epithelial and mesenchymal phenotypes, conferring greater plasticity and potentially enhancing their ability to invade locally or progress. In our case, the observed immunoreactivity appears to correlate with subtypes that are associated with a good prognosis. Similarly, the breast neoplastic cells in our study are undergoing a transition that weakens their epithelial adhesion, but they have not yet fully adopted mesenchymal traits characteristic of an invasive phenotype. This incomplete EMT may explain why these cells have not yet reached full invasiveness. In addition, the references provided on page 8, lines 227-237.
- Comment: Comments on the Quality of English Language. The text added after the first review is difficult to understand.
Response: Agree. Accordingly, I have revised the text to improve its clarity and the quality of the English language.
Changes made in the manuscript are highlighted in bold for clarity.
Reviewer 2 Report
Comments and Suggestions for Authors
Thank you for your continued efforts in revising the manuscript and addressing the concerns raised. I appreciate your dedication to improving the quality of your work. However, after careful consideration, I find that several critical issues remain unresolved, which are essential for supporting your claim of the first reported case of endocrine mucin-producing sweat gland carcinoma (EMPSGC) in a dog. Since authors are presenting this as the first case of EMPSGC in a dog, it is crucial to focus on differentiating EMPSGC from other apocrine gland tumors.
EMPSGC is characterized by three hallmark features:
- Neuroendocrine Differentiation: This requires convincing immunohistochemical evidence using neuroendocrine markers.
- Solid, Papillary, and/or Cribriform Architecture: These architectural patterns should be clearly identifiable histologically.
- Intracellular or Extracellular Mucin: The presence of mucin should be evident through routine or special staining.
Neuroendocrine Differentiation:
While authors have provided synaptophysin staining as evidence of neuroendocrine differentiation, the staining appears relatively weak and may not be sufficient to confidently support the diagnosis of EMPSGC, especially for a first reported case. In comparison to established cases in the literature (e.g., Brett et al., 2017), the intensity and specificity of synaptophysin staining in this cases seem less pronounced. To strengthen your evidence, it is essential to include additional neuroendocrine markers such as chromogranin A. I understand that you faced logistical challenges in obtaining these markers; however, for a definitive diagnosis, especially in a pioneering case, comprehensive immunohistochemical validation is crucial.
Histological Architecture:
Authors have updated Figures 2A and 2B to include appropriate images showing pseudorosettes, cribriform architecture, and cystic formations with mucinous material. However, upon review, the histological features presented do not convincingly demonstrate the characteristic architecture of EMPSGC. For a first case report, it is imperative to provide clear and definitive histological evidence that aligns with the recognized features of EMPSGC, such as well-formed cribriform patterns and pseudorosettes. For your reference, please find link below:https://www.pathologyoutlines.com/topic/skintumornonmelanocyticendocrinemucinsweatglcarcinoma.html
Mucin Detection:
The PAS staining image provided appears to show material consistent with normal apocrine gland secretions rather than true mucin deposition characteristic of EMPSGC. In the absence of abundant extracellular mucin (which, while not always present, is a significant diagnostic feature), the combined lack of strong neuroendocrine marker expression and definitive architectural patterns renders the diagnosis of EMPSGC insufficiently supported.
Given that this is proposed as the first report of EMPSGC in a dog, it is vital to present robust and unequivocal evidence to substantiate the diagnosis. The current data, while suggestive, do not meet the stringent criteria required for such a significant claim.
Additional comment: In Figure 2, authors mention the presence of lymphatic invasion. However, this feature is not convincingly demonstrated in the images provided. For lymphatic invasion to be confirmed, tumor cells should be observed infiltrating within lymphatic vessels that are clearly lined by endothelial cells. The images appear to show an invasive growth pattern or features resembling carcinoma in situ rather than true lymphovascular invasion.
Comments on the Quality of English LanguageThe manuscript contains notable grammatical and structural issues that limit the clarity and readability of the content. Improvements are needed in grammar, sentence structure, and overall coherence to ensure that the scientific findings are communicated effectively. Simplifying complex sentences, using consistent tenses, and correcting article and preposition usage would significantly enhance the quality of the writing.
For example:
Line 51: which have been used for identifying both tumors characteristic -> which have been used to identify tumor characteristics
Line 70: The diagnosis and tumor classification based on distinguished morphological characteristics. -> The diagnosis and tumor classification were made based on distinguishing morphological characteristics.
Line 202: liable possibility-> reliable indicator
Author Response
2nd reviewer
- Comment: Thank you for your continued efforts in revising the manuscript and addressing the concerns raised. I appreciate your dedication to improving the quality of your work. However, after careful consideration, I find that several critical issues remain unresolved, which are essential for supporting your claim of the first reported case of endocrine mucin-producing sweat gland carcinoma (EMPSGC) in a dog. Since authors are presenting this as the first case of EMPSGC in a dog, it is crucial to focus on differentiating EMPSGC from other apocrine gland tumors.
Response: Thank you for pointing this out. I agree with your comment and apologize for the remaining major flaws. We have reviewed our data and strengthened our analysis to address the concern of providing robust evidence for the first report of EMPSGC in a dog. Key updates include emphasizing the role of synaptophysin as an immunohistochemical marker, detailing hallmark histopathological features such as solid architecture and mucin secretion and incorporating comparisons with similar tumors in other species to reinforce our findings. To further address this concern, we performed immunohistochemical staining to differentiate EMPSGC from other apocrine and mammary gland tumors using a combination of markers, including CK19, Sox9, CK5, p63, and vimentin. The markers were selected as they are considered useful in differentiating EMPSGC from apocrine and mammary gland tumors. This point has been added to the conclusion section on page 9, lines 278-280.
- Comment: EMPSGC is characterized by three hallmark features:
- Neuroendocrine Differentiation: This requires convincing immunohistochemical evidence using neuroendocrine markers.
- Solid, Papillary, and/or Cribriform Architecture: These architectural patterns should be clearly identifiable histologically.
- Intracellular or Extracellular Mucin: The presence of mucin should be evident through routine or special staining.
Response: Thank you for pointing this out.
- Neuroendocrine Differentiation: To address this, we performed IHC staining for chromogranin A to evaluate potential neuroendocrine differentiation, which could help distinguish EMPSGC from other apocrine gland tumors. However, chromogranin A was not detected in either case. The absence of chromogranin A expression suggests that these cases lack neuroendocrine features, further supporting their differentiation from neuroendocrine-like or hybrid tumors and ruling out cross-reactivity in the canine species. However, we observed the expression of one neuroendocrine marker, synaptophysin. Therefore, it is considered helpful in distinguishing EMPSGC from these tumors.
- Architecture: Based on three hallmark features, we emphasized the solid architecture and avoided using the cribriform architecture, as it does not convincingly align with the reviewer’s perspective on page 4 lines 143 and 5, lines 165-166.
- Intracellular or Extracellular Mucin: Accordingly, I have updated the PAS staining pictures and designated the areas as a supplementary figure on page 10.
Additionally, the combination of these three main results could help differentiate EMPSGC from other apocrine gland tumors, particularly mammary gland tumors.
- Comment: While authors have provided synaptophysin staining as evidence of neuroendocrine differentiation, the staining appears relatively weak and may not be sufficient to confidently support the diagnosis of EMPSGC, especially for a first reported case. In comparison to established cases in the literature (e.g., Brett et al., 2017), the intensity and specificity of synaptophysin staining in these cases seem less pronounced. To strengthen your evidence, it is essential to include additional neuroendocrine markers such as chromogranin A. I understand that you faced logistical challenges in obtaining these markers; however, for a definitive diagnosis, especially in a pioneering case, comprehensive immunohistochemical validation is crucial.
Response: To address this, we performed IHC staining for chromogranin A to evaluate potential neuroendocrine differentiation, which could help distinguish EMPSGC from mammary or apocrine gland tumors. However, chromogranin A was not detected in our cases (Data not included). The absence of chromogranin A expression suggests that these cases lack neuroendocrine features, further supporting their differentiation from neuroendocrine-like or hybrid tumors and ruling out cross-reactivity in the canine species. However, we observed the expression of one neuroendocrine marker, synaptophysin. Synaptophysin is not expressed in mammary gland tumors or apocrine carcinomas; therefore, it is considered helpful in distinguishing EMPSGC from these tumors. Furthermore, some previous studies reporting synaptophysin expression indicate that it is not a consistently strong neuroendocrine marker, as its expression lacks consistent intensity (Cazzato G, Bellitti E, Trilli I, et al., 2023; Charles NC, Proia AD, Lo C, 2018; Murshed KA, Ben-Gashir M., 2020). This variability in synaptophysin expression highlights its potential utility in supporting the diagnosis of EMPSGC.
- Comment: Histological Architecture: Authors have updated Figures 2A and 2B to include appropriate images showing pseudorosettes, cribriform architecture, and cystic formations with mucinous material. However, upon review, the histological features presented do not convincingly demonstrate the characteristic architecture of EMPSGC. For a first case report, it is imperative to provide clear and definitive histological evidence that aligns with the recognized features of EMPSGC, such as well-formed cribriform patterns and pseudorosettes. For your reference, please find link below:https://www.pathologyoutlines.com/topic/skintumornonmelanocyticendocrinemucinsweatglcarcinoma.html
Response: Thank you for pointing this out. I agree with this comment. Based on our case, we emphasized the solid architecture and avoided using the cribriform architecture, as it does not convincingly align with the reviewer’s perspective. We reviewed and referenced previous studies (Cazzato G, Bellitti E, Trilli I, et al., 2023; Charles NC, Proia AD, Lo C, 2018; Murshed KA, Ben-Gashir M., 2020) that similarly do not conclusively demonstrate the characteristic architecture of EMPSGC. Additionally, we adjusted the wording to describe mucin secretion in cystic areas of the solid architecture and used "pseudoglandular" instead of referring to cribriform patterns and pseudorosettes, as shown in Figures 2A and 2B on page 4, line 143.
- Comment: Mucin Detection:
The PAS staining image provided appears to show material consistent with normal apocrine gland secretions rather than true mucin deposition characteristic of EMPSGC. In the absence of abundant extracellular mucin (which, while not always present, is a significant diagnostic feature), the combined lack of strong neuroendocrine marker expression and definitive architectural patterns renders the diagnosis of EMPSGC insufficiently supported.
Response: Agree. Accordingly, I have updated the PAS staining pictures and designated the areas as a supplementary figure 1 on page 10
10X 40X
- Comment: Given that this is proposed as the first report of EMPSGC in a dog, it is vital to present robust and unequivocal evidence to substantiate the diagnosis. The current data, while suggestive, do not meet the stringent criteria required for such a significant claim.
Response: Thank you for your insightful comment. We acknowledge the importance of providing robust and unequivocal evidence when proposing the first report of EMPSGC in a dog. In response, we have carefully reviewed our data and strengthened our analysis to address this concern.
Immunohistochemical Markers: To substantiate the diagnosis, we have emphasized the significance of the immunohistochemical markers, synaptophysin. This finding aligns with the criteria for diagnosis and differentiation of EMPSGC.
Histopathological Features: We have re-evaluated and detailed the hallmark histopathological features observed in this case, including the solid architecture and mucin secretion patterns, which are consistent with previously described cases of EMPSGC in other species.
Comparison with Previous Studies: We have incorporated additional references and a comparative analysis with similar tumors in other species to reinforce our interpretation and demonstrate that the observed features align with EMPSGC rather than other tumor types.
Clarification of Limitations: We have clearly acknowledged the limitations of our study and the need for further research to validate this as a definitive case of EMPSGC. However, we believe that the combination of histopathological and immunohistochemical findings presented provides compelling evidence for this diagnosis.
We have revised the manuscript to include these clarifications and improvements, which we hope to address your concerns regarding the strength of the evidence. Thank you for highlighting this important aspect, as it has allowed us to enhance the rigor of our work.
- Comment: Additional comment: In Figure 2, authors mention the presence of lymphatic invasion. However, this feature is not convincingly demonstrated in the images provided. For lymphatic invasion to be confirmed, tumor cells should be observed infiltrating within lymphatic vessels that are clearly lined by endothelial cells. The images appear to show an invasive growth pattern or features resembling carcinoma in situ rather than true lymphovascular invasion.
Response: Agree. Accordingly, I have modified the annotation for Figure 2 and removed the mention of lymphovascular invasion on page 5, lines 165-166.
- Comment: Comments on the Quality of English Language
The manuscript contains notable grammatical and structural issues that limit the clarity and readability of the content. Improvements are needed in grammar, sentence structure, and overall coherence to ensure that the scientific findings are communicated effectively. Simplifying complex sentences, using consistent tenses, and correcting article and preposition usage would significantly enhance the quality of the writing.
Response: Accordingly, I have revised the text to improve its clarity and the quality of the English language.
- Comment: For example:
Line 51: which have been used for identifying both tumors characteristic -> which have been used to identify tumor characteristics
Line 70: The diagnosis and tumor classification based on distinguished morphological characteristics. -> The diagnosis and tumor classification were made based on distinguishing morphological characteristics.
Line 202: liable possibility-> reliable indicator
Response: Agree. Accordingly, I have made the modifications as recommended by the reviewer.
- Comment: For example:
Line 51: which have been used for identifying both tumors characteristic -> which have been used to identify tumor characteristics
Line 70: The diagnosis and tumor classification based on distinguished morphological characteristics. -> The diagnosis and tumor classification were made based on distinguishing morphological characteristics
Line 202: liable possibility-> reliable indicator
Response: Agree. Accordingly, I have made the modifications as recommended by the reviewer.
Changes made in the manuscript are highlighted in bold for clarity.
